# River Ecosystem Health Assessment Using a Combination Weighting Method: A Case Study of Beijing Section of Yongding River in China

**DOI:** 10.3390/ijerph192114433

**Published:** 2022-11-04

**Authors:** Linglong Chen, Lan Ma, Jiamen Jiji, Qingqi Kong, Zizhao Ni, Lin Yan, Chengzhong Pan

**Affiliations:** 1Key Laboratory of State Forestry and Grassland Administration on Soil and Water Conservation, School of Soil and Water Conservation, Beijing Forestry University, Beijing 100083, China; 2Beijing Key Laboratory of Urban Hydrological Cycle and Sponge City Technology, College of Water Sciences, Beijing Normal University, Beijing 100875, China

**Keywords:** river ecosystem, health assessment, combination weighting method, Yongding River

## Abstract

(1) Background: River health assessment provides the foundation for sustainable river development and management. However, existing assessments have no uniform standards and methods. (2) Methods: The combination weighting method was proposed, drawing on the advantages of subjective and objective weighting methods. To comprehensively investigate the river health level, an index system based on 16 indices selected from river morphology, river water environment, riparian condition, and social services level was established. The method and framework were applied to the Beijing section of Yongding River in China. (3) Results: The comprehensive weights of river morphology, river water environment, riparian condition, and social services are 0.1614, 0.3170, 0.4459, and 0.0757, respectively. The river health comprehensive index of Yongding River is 3.805; the percentages of excellent, healthy, sub-healthy, unhealthy, and sick river segments are 0%, 11%, 69%, 20%, and 0%, respectively. (4) Conclusions: The results indicate that Yongding River is in a sub-healthy state, and the riparian condition is the key factor that affects the river ecosystem health. Health level exhibited a remarkable spatial variation, mainly influenced by anthropogenic activities, and effective measures are needed to minimize the impact in fragile ecological areas.

## 1. Introduction

Rivers are one of the important ecological corridors, playing a key role in ecosystems and society [1,2]. Rivers provide a range of ecosystem functions, such as material transport, energy cycle, and information exchange between terrestrial and aquatic ecosystems. Socially, rivers accommodate communities by providing a medium for water supply, navigation, fisheries, and cultural recreation. In the process of socio-economic development and anthropogenic activities [3], river ecosystems, especially, are facing unprecedented threats and challenges due to change in river morphology, aquatic habitat, river continuum, water quality, and hydrological characteristics. Simultaneously, river health assessment and impaired river ecosystem restoration aiming at sustainable development are becoming increasingly relevant in the field of river ecology [4,5]. 

To that end, many methods for characterization and assessment of river health have been developed, such as the biological monitoring method [6,7,8], comprehensive index evaluation method [9,10,11], and many mathematical approaches [12,13,14,15]. These methods are not without limitations. For example, the biological monitoring method has a single indicator, which makes it difficult to reflect the all-round changes in river ecosystems. Mathematical methods are frequently limited by sample data. In comparison, the comprehensive index evaluation method is more integrated, making the evaluation results more accurate, rigorous, and reasonable. There are two main types of methods to determine the weight of indices: the subjective assignment method and objective assignment method [12,16]. The subjective weighting method is a method to determine weights based on people's subjective opinions, such as the Delphi method, binomial coefficient method, Analytic Hierarchy Process (AHP), and so on [17,18]. It is superficial but subjective and arbitrary. The objective weighting method can take into account the influence of the actual information of the indices on the evaluation results, mainly principal component analysis [19], entropy method, and so on. It is more suitable for objective reality but may ignore differences in the indices themselves. In general, although many river health evaluation systems have been constructed, since it is challenging to balance subjectivity and objectivity in weight allocation, and indices need to be developed according to the geographical characteristics, climatic conditions, economic development status, and social needs of the evaluation target, there are no unified standards and methods for river health evaluation yet.

Therefore, in this study, we take Yongding River as the research river and select evaluation indices that can reflect the four aspects of river morphology, river water environment, riparian condition, and social service value. A combination weighting method combining the analytic hierarchy process and entropy method was then developed to determine the weight of evaluation indices. Further, the final river health evaluation of Yongding River was carried out by the continuous metrics scoring method. The evaluation results can reflect the river health status of Yongding River comprehensively and provide a theoretical basis for ecological restoration and sustainable management of the river, and also recommend rational and sound reference of river health evaluation methods for other rivers.

## 2. Materials and Methods

An index system based on 16 indices selected from river morphology, river water environment, riparian condition, and social services level has been established to investigate river health level comprehensively. Weight is a value reflecting the importance of the index, and its size dramatically influences the evaluation results. In this study, we integrate subjectivity and objectivity and use the combination of hierarchical analysis and entropy weighting method to determine the weights of evaluation indices. Based on the above methods, the river ecosystem health evaluation system was constructed.

### 2.1. Study Reach and Data Collection

Yongding River originates from Guanshu Mountain in Ningwu County, Shanxi Province. It flows through five districts in Beijing, including Mentougou, Shijingshan, Fengtai, Fangshan, and Daxing. The main channel of the Beijing section of Yongding River is about 189 km long, with a watershed area of about 3200 km^2^. Climatically, the study reach is located in the temperate continental climate zone, with the average annual precipitation about 590 mm, and most of this precipitation occurs during the wet season, from July to September. The terrain of the watershed slopes from northwest to southeast. The upper reaches are dominated by low hills, mountains, and river terraces, while the lower reaches are flat, with vast flatlands on both sides of the river. The soil of study reach is mainly cinnamon soil and fertile, with a thickness of about 20–30 cm, which is suitable for cultivation and production activities. The study reach is located in the section from Zhuwo Reservoir to Wanping Lake in Mentougou District, Beijing, with a total length of about 84 km (Figure 1).

A field survey of the topography, hydrology, and landscape aspects of the rivers in the study reach was conducted in July 2018. Indeed, 175 survey points were distributed from upstream to downstream, following the premise of “one survey point every 500 m, with an additional point in case of rapid changes in ecological conditions”. Taking each 5 survey points as the benchmark and according to the actual situation, the adjacent survey points with similar characteristics are composed of one survey river segment. The study reach was eventually subdivided into 35 river segments, which improved the accuracy of the evaluation results. Data for all indices were obtained from the field surveys.

### 2.2. Evaluation Index System

In order to comprehensively investigate the health level of Yongding River, it is critical to assess the different aspects of river health and use indices to reflect each one. The indices were selected based on physical survey and inspection of the river segments and previous studies [5,20,21,22,23]. According to the context of river health, the index system integrated the ecosystem integrity (river morphology, river water environment, riparian condition) and non-ecological performance (social services) in Yongding River, including target level, criteria level, index level, and 32 indices. River morphology was primarily used to measure the structural features of the river, including the physical structure of the channel, such as riverbed dynamics, riverbed material permeability, water width to river width ratio, and cross-sectional morphology, as well as stream flow morphology, such as longitudinal curvature, sheltered water surface to overall water width ratio, and planar form. River water environment of Yongding River was examined in terms of physical, chemical, and biotic aspects, including physical factors (odor, water temperature, water turbidity, oxidation-reduction potential (ORP), flow rate ratio, total dissolved solids (TDS)), chemical factors (ammonia nitrogen (NH_3_-N), phosphate, chemical oxygen demand (COD), total phosphorus (TP), dissolved oxygen (DO), pH), and biological factors (benthic habitat conditions). The riparian zone is a transitional zone extending from river to land, and its structure and function play an important role in river ecological protection. Riparian condition indices are used to reveal the structural and functional status of the riparian zone, including land use, the number of water conservancy projects, riparian structure, erosion degree, slope, vegetation cover, vegetation width, structural integrity, vegetation diversity, and riparian zone accessibility. The indices reflecting social service functions contained landscape diversity index and ornamental and recreation value.

Based on previous studies [24,25], the above candidate indices were further screened. Principal component analysis (PCA) was undertaken using SPSS 26, and the indices with scores >0.6 in each PC axis were considered the key indices that play a dominant role in river health. For the remaining metrics, boxplots and Spearman correlation analysis were used to examine the independence among the indices. Metrics were sequentially evaluated for redundancy using correlation analysis, and those with correlation coefficients |r| > 0.75 and *p* < 0.01 were screened out, and, finally, only one index with greater discriminatory ability (within each pair of strongly correlated indices) was retained based on its performance in the box plot [26,27].

Based on the above screening, the evaluation index system was finally constructed (Table 1). Within the four metric categories, namely the river morphology, river water environment, riparian condition, and social services that describe the river health condition, a total of 16 qualitative and quantitative indices were selected.

### 2.3. Determination of Index Weights

#### 2.3.1. AHP

First, the relative importance of the river health evaluation indices was compared and scored by experts, and the judgment matrix was constructed according to the 1–9 scale proposed by Professor Saaty [28]:(1)$$A={\left(a_{ij}\right)}_{m\times n}\;,\;i=1,\;2,\;\ldots ,\;m;\;j=1,\;2,\;\ldots ,\;n,\;a_{ij}>0;\;a_{ij}=1/a_{ji};\;a_{ij}=1.$$
where $a_{ij}$ is the ratio of the importance of index i to index j.

The weight vector was then measured as follows:(2)$$\lambda_{\max}=\overset{n}{\underset{i=1}{\sum }}\frac{\left(A\alpha_{i}\right)}{n\alpha_{i}}$$
where $\lambda_{\max}$ is the maximum feature vector, and $\alpha_{i}$ is the weight vector.

After obtaining, the consistency test of the judgment matrix is required. Detailed procedure was as follows [29]:(3)$$CI=\frac{\lambda_{\max}-n}{n-1}$$
(4)$$CR=\frac{CI}{RI}$$
where n is the order of the judgment matrix, CI is the consistency index, RI is the random consistency index of the judgment matrix. When CR < 0.1, the judgment matrix is considered to pass the consistency test.

#### 2.3.2. Entropy Weight Method

Primarily, samples and indices were selected and defined as m samples and n indices, and the indices were normalized to obtain the data matrix:(5)$$X={\left(x_{ij}\right)}_{m\times n}$$
where $x_{ij}$ is the original value of sample *i*, index *j*.

According to the definition, the entropy value of the *j*th term was calculated as follows:(6)$$e_{j}=-\sum_{n=1}^{m}P_{ij}lnP_{ij},\;i=1,\;2,\;\ldots ,\;m;\;j=1,\;2,\;\ldots ,\;n.$$

The weight of the *j*th index was calculated as follows:(7)$$d_{j}=\frac{1}{e_{j}}\;,\;j=1,\;2,\;\ldots ,\;n.$$
where $d_{j}$ was normalized to obtain the weight of the *j*th index. Specific calculation formula is as follows:(8)$$W_{j}=\frac{d_{j}}{\sum_{j=1}^{n}d_{j}}$$

#### 2.3.3. Combination Weighting Method

Assuming the weight vector calculated by the hierarchical analysis is $\alpha =\left(\alpha_{1},\alpha_{2},\ldots ,\alpha_{n}\right)$ and the weight vector calculated by the entropy weight method is $\beta =\left(\beta_{1},\beta_{2},\ldots ,\beta_{n}\right)$.

Combined weights were calculated based on geometric average method, and the calculation formula was as follows:(9)$$W_{j}=\frac{\sqrt{\alpha_{i}\beta_{j}}}{\sum_{j=1}^{n}\sqrt{\alpha_{i}\beta_{j}}},\;j=1,\;2,\;\ldots ,\;n.$$

### 2.4. Evaluation Standards

In order to improve accuracy, in this paper, the quartile method was used to score, and the evaluation criteria are shown in Table 2. The maximum value, minimum value, mean value, and standard deviation of each index were counted. The five quartiles of 5%, 25%, 50%, 75%, and 95% of each index were calculated to establish the river health evaluation criteria. The indices were assigned scores according to the constructed evaluation criteria. According to the total score of indices in descending order, the river health status is divided into five levels, namely excellent, healthy, sub-healthy, unhealthy, and sick. The within-average linkage method and squared Euclidean distance were used for cluster analysis of evaluation results.

According to the constructed river health evaluation index system, the weighted average method is used to calculate the comprehensive river health index. The specific calculation process was as follows:(10)$$RHI=\sum_{i=1}^{n}H_{i}W_{i}$$
where *RHI* is the river health comprehensive index, $H_{i}$ is the score of the ith index, and $W_{i}$ is the comprehensive weight of the *i*th index.

## 3. Results

### 3.1. Weighting of Evaluation Indices

The weight is calculated according to the analytic hierarchy process and entropy weight method, respectively, and the comprehensive weight is computed. The results are shown in Table 3. The comprehensive weights of river morphology, river water environment, riparian condition, and social services in the criterion level are 0.1614, 0.3170, 0.4459, and 0.0757, respectively. Moreover, it suggests that riparian condition has the greatest impact on river health. The value range of the index level is 0.0072–0.2067, where the weights of riparian vegetation width and TDS are larger and the weights of odor, riparian vegetation cover, and riparian erosion degree are smaller.

### 3.2. Evaluation Criteria

According to the statistics of the maximum value, minimum value, and average value of each index, and calculating the five quantiles of 5%, 25%, 50%, 75%, and 95% of each index, the index scoring standard of the Beijing section of Yongding River is obtained (Table 4).

The evaluation index system of Yongding River consists of one first-class index (target level), four second-class indices (criterion level), and sixteen third-class indices (index level). Each index has the highest score of 8 and the lowest score of 0. According to the calculated river health comprehensive index value, the total score is divided into five equal points to construct the river evaluation criteria. The comprehensive evaluation grade of Yongding River as divided is shown in Table 5.

### 3.3. Evaluation Results

#### 3.3.1. Evaluation Results of the Criterion Level

The statistics on the health status of the criterion level of Yongding River are shown in Table 6. In general, the river morphology, river water environment condition, riparian condition, and social services of the whole river reach are generally in a sub-healthy state, and the health composite indices are 3.360, 4.616, 3.260, and 4.571, respectively. In terms of river morphology, the health index of each river segment ranged from 0.729 to 6.159, and unhealthy segments account for as high as 37%. In terms of river water environment, the health index of each river segment ranged from 2.72 to 7.362. The proportion of river segments whose evaluation level is not lower than sub-healthy is 86%, so the overall morphological conditions of rivers were good. Regarding riparian conditions, the health composite index ranges from 0.878 to 6.210. Furthermore, the percentages of unhealthy segments are as high as 46%, indicating that the riparian condition of the study reach needs to be improved. In terms of social services, its health composite index has the most considerable difference, with a standard deviation of 2.321, indicating that landscape diversity varies widely among river segments.

The hierarchical clustering results of river health evaluation are shown in Table 7. River segments S3, S4, S8, S9, S20, and S22 showed poor river morphology and riparian condition, while river water environment and social services were above sub-healthy level. Among them, the river morphology and riparian condition indices of river segment S8 are the lowest values in the whole river, with a difference of 4.760 and 4.113 from the highest value (6.159 and 6.210), with a significant difference of 79% and 75%, respectively. River segments S5, S10, S13, and S18 only have riparian conditions that do not reach the sub-healthy level. For these river segments, the management and improvement of riparian should be emphasized in ecological restoration work. While river segments S15 and S27 only have unhealthy social services, the rest perform well. The health indices of river morphology and riparian condition of S15 are 6.159 and 6.210, respectively, the highest values of the whole river. For such river segments, emphasis can be placed on improving the value of social services while maintaining the existing condition.

#### 3.3.2. Comprehensive Evaluation Results

The results of the health level of Yongding River are shown in Figure 2. Overall, the health composite index of the study reach of Yongding River is 3.805, which is at the sub-healthy level. The percentages of excellent, healthy, sub-healthy, unhealthy, and sick river segments are 0%, 11%, 69%, 20%, and 0%, respectively. Among them, the comprehensive health index of river segment S16 reached 5.632, which is the optimal value; the comprehensive health index of river section S30 was only 2.707 as the lowest value, and the degree of difference from the optimal value reached 52%.

Combined with the analysis of the field survey results (Table 8), the evaluation level of river segments S15, S16, S17, and S27 is at a healthy state. Among them, river segments S16 and S17 are close to Wangping Town. In recent years, the government has continued to carry out particular rectification actions along Yongding River and has removed open-air barbecue stalls and illegal construction areas of more than 23,400 m^2^, with good results of comprehensive river management. The river water body is clear and odorless. The river morphology and riparian conditions are in a healthy state. In contrast, river segments S15 and S27 have a good water environment near the mountains because they are far away from residential areas and have less human interference. The riparian trees, shrubs, and herbaceous vegetation hierarchy are well-defined, and the vegetation cover is high. The vegetation diversity in the buffer zone is greater than 1.1829 and higher than 80% of river segments, which is higher than the average value of Yongding River.

Twenty-four river segments are in sub-healthy condition, among which S1, S3, S4, S5, S6, and S7 are located in the zone of the ecological project “Five Lakes and One Line”. After implementation of the water purification project, river ecological restoration project, and construction of large country parks, the area has high riparian vegetation coverage. The vegetation diversity index is higher than average, and the river has better water environment conditions and high ornamental and recreational value. However, the river banks are mostly protected by grouted rubble, dry rubble, and lead wire gabions built in the river channel project, making the channel planform straight. It not only fails to provide a natural habitat for life below water but forces the water flow faster than its natural form, leaving less time for micro-organisms to decompose pollutants in the water and not reducing accelerated erosion caused by anthropogenic factors. S9, S12, S14, S23, S25, S28, and S33 are amid many tourist attractions and farmhouses, with high ornamental and recreational value but under high interference of human activities. For example, along S9, farmhouses, convenience stores, and restaurants have been densely constructed. Along S23, the scenic spots of Eighteen Ponds attract millions of tourists, mainly from the urban area, every holiday. In these areas, due to the waste produced by tourists and construction, water bodies are polluted so that turbidity is high, accompanied by a worsening benthic habitat; vegetation coverage is also lower than average, with vegetation diversity indices of S12, S14, S23, S25, and S33, and riparian zones are 0.8887, 0.9835, 0.7954, 0.5679, and 0.8509, respectively, which are lower than the average value. S10 and S35 are more affected by construction of water conservancy projects; S35 is adjacent to Sanjiadian Reservoir and Zhuwo Reservoir, respectively, and a large number of irrigation diversions, large power plants, and barrage dams are built in river segment S10. The land use on both sides of these river segments is mostly for construction and roads, and the vegetation widths of riparian zones in river segments S10 and S35 are less than 8 m and 5 m, respectively, which is lower than 77% of the river sections. The vegetation diversity index of the riparian zone in river segment S35 is only 0.4283, which is 56% different from average.

There are seven river segments at an unhealthy level. River segment S8 is adjacent to Sanjiadian Reservoir, with a primary flood control function but a weakened ecological function. Both river banks are slurry stone revetment, and riverside wetlands and buffer zones have disappeared. The vegetation on both sides of the river has been heavily damaged, and the vegetation diversity index is only 0.5475, which is much lower than the average value of Yongding River. River section S21 is close to Anjiazhuang, and the land use type is primarily residential and construction land, with low ornamental recreation value. The water body is relatively turbid, with high ammonia nitrogen content, and the river water environment conditions are poor. River segment S30 is located in the vicinity of Qingbaikou Village, and the surrounding farmland is extensive. The diversity of vegetation has been significantly damaged, especially at the junction of the land and water counterparts.

## 4. Discussion

### 4.1. Evaluation Method Reliability Analysis

There are many methods for river health evaluation, and calculation of index weights is critical. Different weight calculation methods may produce different evaluation results. In order to synthesize the advantages of the subjective and objective weighting method, in this paper, we use the combined assignment method of hierarchical analysis and entropy method to determine the index weights. The core problem of combination weighting is determining the weight distribution of the two methods. There are many studies in this area [30,31,32]. However, the mathematical derivation of most methods is cumbersome, poorly applied, and not operable. Accurately carrying out combination weighting now seems to have no suitable method. Therefore, to simplify the evaluation system, we used the geometric mean method for combination assignment.

The weighting results of evaluation indices are riparian condition index > river water environment index > river morphology index > river social service function index. Based on the field survey analysis and evaluation results, the main influencing factor of the health status of Yongding River is the riparian condition. It may be because the Yongding River bank is most affected by human activities, such as construction of illegal buildings on the riverbank and tourism development. Moreover, the information provided by riparian condition indices is more decadent. Therefore, more attention should be paid to riparian condition indices in future studies.

In addition, the evaluation system used in this study is based on the Beijing section of Yongding River, which has some shortcomings. For example, the influence of time dynamics on the river evaluation index is not considered. Moreover, the selection of evaluation indicators is too subjective. Aquatic life indicators, such as phytoplankton, and other socio-economic indices are not considered due to the limitations of conditions and data. Therefore, for rivers with different spatial scales and regional differences, the river health evaluation system should be further screened and judged according to the actual situation of rivers. Moreover, the method tends to lack consideration of information redundancy among the various indicators. Long-term use of the indicator systems should be further studied.

Overall, we have established a health evaluation index system for Yongding River and proposed restoration and management countermeasures for the current health status of Yongding River. This study broadens the ideas and methods of river health evaluation research, and the results can provide decision-making reference for health assessment, management, and protection of other rivers.

### 4.2. Suggestions on River Ecological Restoration

The health condition of the Beijing section of Yongding River shows a series of problems, such as water environment pollution, riparian ecological degradation, urbanization, and severe human impact. The manner to restore the damaged natural environmental systems is one of the essential issues that river managers and the public should consider. In this paper, we propose the following recommendations for restoring rivers in different health conditions.

For healthy river segments, river management should be strengthened to maintain the morphology, riparian condition, and water environment quality while increasing the value of social services. For those in a sub-healthy state, targeted restoration should be carried out. For river segments affected by land use changing engineering construction, decision-makers should have a clear picture of the impact of engineering construction and irrigation diversions on the river morphology and river water environment. More stringent measures to regulate illegal farming or urban expansion should be introduced. For river segments where the habitat is damaged by landscape recreation, tourism management should be regulated, and discharge of pollutants from production and construction and disposal of household waste should be monitored to reduce water pollution; at the same time, ecological protection, restoration, remediation, and reconstruction in riparian zones should be carried out, building ecological slopes with trees, shrubs, and herbage. For unhealthy river segments, comprehensive treatment should be carried out for all aspects: regularly conducting river training to reduce the pollutant precipitated on the riverbed, optimizing riparian ecological construction, focusing on improving river morphology and riparian condition, restoring natural river channel state, and improving the stability and biodiversity of riparian zones.

## 5. Conclusions

In this research, we constructed a river health evaluation system based on 16 indices in four aspects: river morphology, river water environment, riparian condition, and social service value, and evaluated the health status of Yongding River:(1)The river health evaluation indices were assigned using a combination of hierarchical analysis and entropy weighting methods. The weight was ranked as riparian condition > river water environment > river morphology > social service function. It shows that the river water environment has the most significant influence on the health status of the Beijing section of Yongding River.(2)The evaluation results using the continuous metrics scoring method show that the overall health index for the study reach of Yongding River is 3.805, which is at a sub-healthy level. For the 35 segments, the percentages of excellent, healthy, sub-healthy, unhealthy, and sick river segments are 0%, 11%, 69%, 20%, and 0%, respectively.(3)Combining the actual situation of Yongding River, factors affecting the health of Yongding River include mainly human disturbance activities, such as industrial and agricultural development, construction of water conservancy projects, tourism and sightseeing, and so forth. In order to improve the health condition, Yongding River channel and riparian regulation need to be strengthened.

## Figures and Tables

**Figure 1 ijerph-19-14433-f001:**
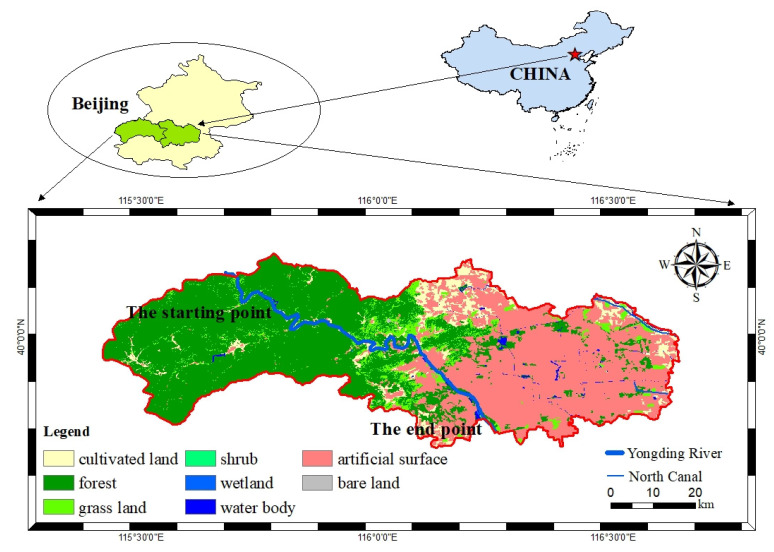
The location of the study reach.

**Figure 2 ijerph-19-14433-f002:**
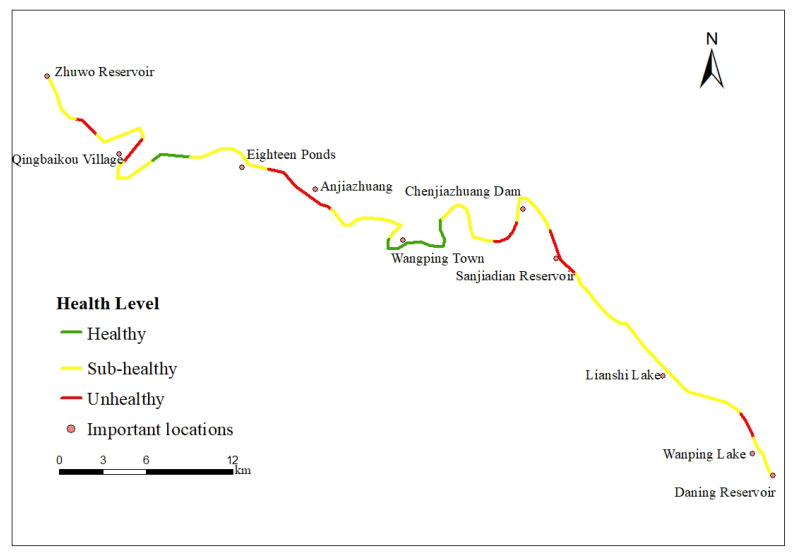
Comprehensive health evaluation results of Yongding River.

**Table 1 ijerph-19-14433-t001:** Index system of health assessment for Yongding River.

Target Level	Criterion Level	Index Level	Basic Description	Calculation Method/Data Resource
River health	River morphology (B1)	Planar morphology (C1)	The degree of meandering rivers	The number of sharp bends and river islands
		Riverbed material permeability (C2)	The permeability of riverbed materials	Qualitative description, assigning a score of 1–4 from low to high according to the degree of water permeability
		Water width to river width ratio (C3)	Water width as a percentage of overall river width	Measurement with portable range finder
		Sheltered water surface to overall water width ratio (C4)	Proportion of water surface with shade to the whole water surface	Measurement with portable range finder
	River water environment (B2)	Odor (C5)	Describe whether the river water has a fishy smell	Qualitative description, assigning a score of 1–4 from low to high according to the degree of odor emitted
		Flow rate ratio (C6)	Ratio of maximum to minimum river flow velocity	Measurement with a flow meter
		TDS(μs/cm) (C7)	Conductivity of the river	Measurement by handheld conductivity meter
		TP (C8)	Total phosphorus content of the river	Measurement by Multi-parameter Water Quality Analyzer
		DO (C9)	Dissolved oxygen content of the river	Measurement with Seven2Go Pro S9 portable dissolved oxygen meter
	Riparian condition (B3)	erosion degree (C10)	Extent of riparian erosion	Qualitative description, assigning a score of 1–4 from low to high depending on the degree of erosion
		vegetation cover (C11)	Percentage of vegetation zones on riparian zones	Qualitative description, assigning a score of 1–4 from lowest to highest according to the percentage of vegetation zones
		vegetation diversity (C12)	Diversity level of riparian vegetation species	Calculation of Shannon–Wiener Diversity Index
		vegetation width/m (C13)	Width of riparian vegetation zone	Measurement with portable range finder
		Slope (C14)	Slope of the riparian zone	Measurement by slope meter
		Water conservancy projects(C15)	The extent of construction of hydraulic engineering measures in the riparian zone that affect water flow	Number of artificial engineering measures affecting water flow
	Social services (B4)	Ornamental and recreation value (C16)	The level of recreational value provided by the river system	Calculation with Romme Landscape Richness Index

**Table 2 ijerph-19-14433-t002:** Health assessment grading of river ecosystem.

Quantile	≥95%	75–95%	50–75%	25–50%	<25%
Standard	8	6	4	2	0
Health level	Excellent	Healthy	Sub-healthy	Unhealthy	Morbid

**Table 3 ijerph-19-14433-t003:** The weight of every river health assessment index.

Index	River Morphology (B1)				River Water Environment (B2)					Riparian Condition (B3)						Social Services (B4)
	C1	C2	C3	C4	C5	C6	C7	C8	C9	C10	C11	C12	C13	C14	C15	C16
Weight obtained by AHP	0.0406	0.0575	0.0218	0.0141	0.1354	0.0487	0.0704	0.0869	0.1239	0.0265	0.0354	0.075	0.0582	0.0439	0.0882	0.0736
Weight obtained by entropy method	0.0371	0.0237	0.1409	0.1364	0.0026	0.0942	0.0879	0.0294	0.0142	0.0257	0.0247	0.0374	0.1728	0.0809	0.0424	0.0501
Comprehensive weight (*W_i_*)	0.0309	0.028	0.0631	0.0395	0.0072	0.0949	0.1264	0.0525	0.0361	0.014	0.018	0.0576	0.2067	0.0729	0.0768	0.0757

**Table 4 ijerph-19-14433-t004:** Index evaluation criteria.

Index		Quantile		Score Standard				
		5%	95%	8	6	4	2	0
River morphology (B1)	C1	0.25	1	≥1.000	0.750–1.000	0.500–0.750	0.250–0.500	<0.250
	C2	0.25	1	≥1.000	0.750–1.000	0.500–0.750	0.250–0.500	<0.250
	C3	0.013	0.862	≥0.862	0.421–0.862	0.182–0.421	0.093–0.182	<0.093
	C4	0.02	0.748	≥0.748	0.281–0.748	0.184–0.281	0.103–0.184	<0.103
River water environment (B2)	C5	0.25	1	≥1.000	0.750–1.000	0.500–0.750	0.250–0.500	<0.250
	C6	0.056	0.899	≥0.899	0.506–0.899	0.337–0.506	0.174–0.337	<0.174
	C7	0.048	0.82	≥0.820	0.424–0.820	0.337–0.424	0.163–0.337	<0.163
	C8	0.333	0.833	≥0.833	0.833–1.000	0.667–0.833	0.500–0.667	<0.500
	C9	0.585	0.94	≥0.940	0.849–0.940	0.806–0.849	0.764–0.806	<0.764
Riparian condition (B3)	C10	0.25	1	≥1.000	0.750–1.000	0.500–0.750	0.250–0.500	<0.250
	C11	0.25	1	≥1.000	0.750–1.000	0.500–0.750	0.250–0.500	<0.250
	C12	0.153	0.948	≥0.948	0.827–0.948	0.665–0.827	0.484–0.665	<0.484
	C13	0.022	0.336	≥0.336	0.141–0.336	0.090–0.141	0.055–0.090	<0.055
	C14	0.025	0.916	≥0.916	0.750–0.916	0.484–0.750	0.282–0.484	<0.282
	C15	0.16	0.953	≥0.953	0.817–0.953	0.667–0.817	0.467–0.667	<0.467
Social services (B4)	C16	0.25	1	≥1.000	0.750–1.000	0.500–0.750	0.250–0.500	<0.250

**Table 5 ijerph-19-14433-t005:** River health assessment criteria of Yongding River.

Health Level	Excellent	Healthy	Sub-Healthy	Unhealthy	Sick
RHI Value	8.0–6.4	4.8–6.4	3.2–4.8	1.6–3.2	0–1.6

**Table 6 ijerph-19-14433-t006:** Statistics on the health status of criterion level of Yongding River.

Health Level	River Morphology	River Water Environment	Riparian Condition	Social Service
Excellent	0%	6%	0%	20%
Healthy	23%	40%	17%	26%
Sub-healthy	26%	40%	29%	17%
Unhealthy	37%	14%	46%	37%
Sick	14%	0%	9%	0%

**Table 7 ijerph-19-14433-t007:** Data description of each cluster of hierarchical cluster analysis based on evaluation results of the criterion level of Yongding River.

Cluster	Segments	Criterion Level	Mean	Skewness	Kurtosis
1	9 (S1, S2, S3, S4, S8, S9, S20, S22, S23)	River_morphology	2.162 ± 0.916	−0.285 ± 0.661	−1.312 ± 1.279
		River_water_environment	4.735 ± 1.066	−0.415 ± 0.661	−0.189 ± 1.279
		River_bank_condition	2.367 ± 1.012	0.537 ± 0.661	0.068 ± 1.279
		Social_service	7.091 ± 1.044	−0.213 ± 0.661	−2.444 ± 1.279
2	12 (S5, S10, S13, S14, S16, S17, S18, S19, S30, S33, S34, S35)	River_morphology	4.663 ± 0.504	0.201 ± 0.913	0.578 ± 2.000
		River_water_environment	4.817 ± 1.120	−0.489 ± 0.913	1.815 ± 2.000
		River_bank_condition	3.414 ± 1.523	0.224 ± 0.913	−2.207 ± 2.000
		Social_service	6.400 ± 0.894	2.236 ± 0.913	5.000 ± 2.000
3	9 (S6, S7, S15, S21, S24, S25, S26, S27, S29)	River_morphology	2.767 ± 1.015	0.807 ± 0.661	0.61 ± 1.279
		River_water_environment	4.056 ± 0.924	0.644 ± 0.661	−0.804 ± 1.279
		River_bank_condition	3.130 ± 0.965	0.494 ± 0.661	−1.036 ± 1.279
		Social_service	2.909 ± 1.044	0.213 ± 0.661	−2.444 ± 1.279
4	2 (S11, S12)	River_morphology	4.824 ± 0.817	1.330 ± 0.913	2.307 ± 2.000
		River_water_environment	4.191 ± 1.068	0.114 ± 0.913	−1.465 ± 2.000
		River_bank_condition	5.103 ± 0.899	0.125 ± 0.913	−2.351 ± 2.000
		Social_service	2.400 ± 0.894	2.236 ± 0.913	5.000 ± 2.000
5	3 (S28, S31, S32)	River_morphology	5.312 ± 0.734	1.732 ± 1.225	-
		River_water_environment	6.604 ± 0.782	−0.263 ± 1.225	-
		River_bank_condition	3.68 ± 1.392	1.728 ± 1.225	-
		Social_service	2.000 ± 0	-	-

**Table 8 ijerph-19-14433-t008:** Data of main indices of Yongding River.

Health Level	River Segment Number	Flow Rate Ratio	TDS (μs/ cm)	TP (mg/L)	DO (mg/L)	Riparian Zone Vegetation Diversity	Riparian Slope (°)	Riparian Vegetation Width (m)	Ornamental and Recreation Value	Erosion Degree	Vegetation Cover
Healthy	S16	9.588	1417	0.02	9.45	0.9555	54.6	154	1	1	3
	S17	6.520	1495	0.01	10.62	1.1955	38.4	34	2	1	2
	S15	1.746	1467	0.03	4.77	1.1829	51.4	56	1	2	3
	S27	4.952	1498	0.01	10.03	1.3015	10.4	30	1	2	2
Sub-healthy	S28	1.000	1495	0.01	10.18	1.0696	17.4	50	4	2	3
	S31	4.110	1496	0.01	9.99	1.1629	69.2	18	3	3	4
	S18	6.210	1461	0.01	10.75	1.1283	55.6	10	1	1	4
	S32	3.894	1483	0.01	10.08	1.0113	73.6	12	3	3	4
	S13	2.676	1473	0.02	10.44	1.1371	18.8	14	1	1	2
	S14	6.271	1469	0.03	10.50	0.9835	41.0	18	1	1	1
	S19	1.532	1466	0.01	10.48	0.8510	60.2	28	2	2	2
	S23	3.496	1474	0.01	10.31	0.7954	26.2	18	3	2	1
	S26	3.490	1490	0.04	9.83	1.2218	19.0	23	1	2	2
	S33	3.246	1494	0.02	10.71	0.8509	49.4	18	2	2	2
	S7	1.531	1476	0.01	7.34	1.0084	78.0	26	3	1	4
	S29	1.679	1490	0.02	10.38	0.9842	33.8	16	1	2	2
	S1	1.359	1429	0.01	10.67	1.1247	43.5	9	3	2	4
	S10	4.961	1468	0.03	10.25	0.9825	20.6	8	3	1	2
	S25	4.497	1485	0.01	9.63	0.5679	26.2	29	1	1	2
	S12	4.235	1472	0.01	11.26	0.8887	44.0	16	2	1	1
	S24	3.420	1483	0.01	9.76	0.6623	20.0	14	2	1	2
Unhealthy	S11	4.725	1475	0.00	11.69	1.1663	68.2	13	1	2	2
	S22	2.560	1472	0.02	10.18	0.8004	11.8	12	3	2	2
	S34	3.487	1501	0.03	10.54	0.6623	72.0	10	1	2	3
	S2	2.195	1445	0.02	10.42	0.7872	69.2	11	4	2	2
	S21	1.588	1482	0.06	10.09	0.9959	29.4	20	1	2	2
	S30	2.435	1488	0.01	10.35	0.8610	57.0	14	1	2	2
Average value		4.191	1474	0.02	10.15	0.9856	42.6	20	2	1	1

## Data Availability

The data presented in this study are available on request from the corresponding author.

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
