# Peer review of "River Ecosystem Health Assessment Using a Combination Weighting Method: A Case Study of Beijing Section of Yongding River in China"

_ijerph, 2022, doi:10.3390/ijerph192114433_

Round 1

Reviewer 1 Report

An interesting method for assessing river ecosystem health is presented and trialled in a river in China.

The principal methodological references appear to be mainly drawn from China, and it would be good to refer to other river health assessment methods, e.g. the Freshwater Health Index [1]developed through Conservation International[2] [3]and the chemical and biotic indices as used by many river management agencies e.g. Mekong River Commission[4], so that the reader can see how this method fits in within the global family of river health indices[5] [6]. It is also important to make it clear that this method is based upon physical survey and inspection of the river segments.

Language issues. The paper is written in a slightly condensed way and sometimes the choice of words is a rather curious, making the reading flow a bit difficult. I suspect that this results from Google translate word selection, e.g. use of connotation in line 90 may be better as context. So a good English edit would assist the readers understanding.

Figures. Both figures need to be a large size – at least full page width, with clear legends, scales etc. Figure 1 might be better with landuse legend rather than elevation, and Figure 2 should also show clearly the segment numbers the landscape features mentioned in the text, e.g. Wangping Town, the Five Lakes and one line project, eighteen ponds, the reservoirs and the power plants and dams. This would help to illustrate and provide a rationale for why the segments are more or less healthy.

At the front of the methodology section it would be good to provide an overview of the process or flow diagram and to explain why each step is being carried out – why it is necessary for a better combination index.

Table 1 would benefit from an additional  column showing the range of values expected and the scoring process used, especially for the more qualitative calculation methods. Please clarify whether flow rate is the cubic metres per sec flow or the velocity of the water in m/sec. If greater clarity is required a table for the index measurements for each segment could be included in the supplementary materials.

Table 3 – and Table 4 – it would be clearer to group the indices ID with their respective criterion levels. In table 3, the numbers of the weightings do not bear any relationship to the numbers mentioned in the text, making it difficult to relate the two.

Table 4 – Typo in the headers – Score Standard. The  table and text needs to make it clear that the table provides the scoring system for the ranges of combination weights for each index. Possibly the table could be reformatted to show this, it is not immediately apparent.

Table 8 – deals with the quantitative indices, it would be useful to have a similar table explaining how the more qualitative indices have been used and incorporated into the scoring system, otherwise it might be concluded that these are less important than the quantitative.

The conclusion in line 301 that the riparian condition index reflects the river health more than river water environment index more than  river morphology more than social service index could be explored a bit more, explaining why this might be and leading to a conclusion that a shortened version of this method could focus on riparian condition alone?

[1] Vollmer, D.S.K.; Souter, N.J.; Farrell, T.; Dudgeon, D.; Sullivan, C.A.; Fauconnier, I.; MacDonald, G.M.; McCartney, M.P.; Power, A.G.; McNally, A.; et al. Integrating the social, hydrological and ecological dimensions of freshwater health: The Freshwater Health Index. Sci. Total Environ. 2018, 627, 304–313, doi:10.1016/j.scitotenv.2018.01.040.

[2] Shaad, K., Souter, N.J., Vollmer, D. et al. Integrating Ecosystem Services Into Water Resource Management: An Indicator-Based Approach. Environmental Management 69, 752–767 (2022). https://doi.org/10.1007/s00267-021-01559-7

https://www.freshwaterhealthindex.org/fact-sheets

[3] Souter, N.J.; Shaad, K.; Vollmer, D.; Regan, H.M.; Farrell, T.A.; Arnaiz, M.; Meynell, P.-J.; Cochrane, T.A.; Arias, M.E.; Piman, T.; Andelman, S.J. Using the Freshwater Health Index to Assess Hydropower Development Scenarios in the Sesan, Srepok and Sekong River Basin. Water 202012, 788. https://doi.org/10.3390/w12030788

[4] https://www.mrcmekong.org/our-work/functions/basin-monitoring/ and

https://www.mrcmekong.org/our-work/functions/basin-monitoring/ecological-health-monitoring/

[5] Böck, K.; Polt, R.; Schülting, L. Ecosystem Services in River Landscapes. In Riverine Ecosystem Management; Schmutz, S., Sendzimir, J., Eds.; Springer: Cham, Switzerland, 2018; Volume 8, pp. 409–434.

[6] European Commission. Common Implementation Strategy for the Water Framework Directive (2000/60/EC): Guidance Document No 13. Overall Approach to the Classification of Ecological Status and Ecological Potential; European Commission: Brussels, Belgium, 2005.

DPIPWE. Summary of the CFEV Assessment Framework. v1.3. Conservation of Freshwater Ecosystem Values Program; Department of Primary Industries, Parks, Water and Environment: Hobart, Tasmania, Australia, 2014.

Nel, J.; Colvin, C.; Le Maitre, D.C.; Smith, J.; Haines, I. Defining South Africa’s Water Source Areas; WWF-World Wide Fund for Nature: Cape Town, South Africa, 2013.

Nel, J.L.; Driver, A.; Strydom, W.F.; Maherry, A.; Petersen, C.; Hill, L.; Roux, D.J.; Nienaber, S.; van Deventer, H.; Swartz, E.; et al. Atlas of Freshwater Ecosystem Priority Areas in South Africa: Maps to Support Sustainable Development of Water Resources; Water Research Commission: Gezina, South Africa, 2011.

Author Response

Response to Reviewer 1 Comments

Point 1: The principal methodological references appear to be mainly drawn from China, and it would be good to refer to other river health assessment methods, e.g. the Freshwater Health Index [1]developed through Conservation International[2] [3]and the chemical and biotic indices as used by many river management agencies e.g. Mekong River Commission[4], so that the reader can see how this method fits in within the global family of river health indices[5] [6]. It is also important to make it clear that this method is based upon physical survey and inspection of the river segments.

Response 1: We have reorganized the introductory section, referring to the literature you provided.

Point 2: Language issues. The paper is written in a slightly condensed way and sometimes the choice of words is a rather curious, making the reading flow a bit difficult. I suspect that this results from Google translate word selection, e.g. use of connotation in line 90 may be better as context. So a good English edit would assist the readers understanding.

Response 2: Language has been refined throughout the text.

Point 3: Figures. Both figures need to be a large size – at least full page width, with clear legends, scales etc. Figure 1 might be better with landuse legend rather than elevation, and Figure 2 should also show clearly the segment numbers the landscape features mentioned in the text, e.g. Wangping Town, the Five Lakes and one line project, eighteen ponds, the reservoirs and the power plants and dams. This would help to illustrate and provide a rationale for why the segments are more or less healthy.

Response 3: Improvements have been made to both figures, the legend of Figure 1 has been changed to land use, and the segment numbers the landscape features mentioned in the text have been added to Figure 2.

Point 4: At the front of the methodology section it would be good to provide an overview of the process or flow diagram and to explain why each step is being carried out – why it is necessary for a better combination index.

Response 4: We have added an overview of the process earlier in the methodology section, explaining why each step is being carried out and why it is necessary for a better combination index.

Point 5: Table 1 would benefit from an additional  column showing the range of values expected and the scoring process used, especially for the more qualitative calculation methods. Please clarify whether flow rate is the cubic metres per sec flow or the velocity of the water in m/sec. If greater clarity is required a table for the index measurements for each segment could be included in the supplementary materials.

Response 5: The specific scoring process and the range of values are shown in Table 4, and in Table 1 we add a description of the qualitative metrics and the data sources. In this paper, the flow rate ratio is used instead of the flow rate, which is a unitless metric.

Point 6: Table 3 – and Table 4 – it would be clearer to group the indices ID with their respective criterion levels. In table 3, the numbers of the weightings do not bear any relationship to the numbers mentioned in the text, making it difficult to relate the two.

Response 6: We have grouped Tables 3 and 4 by criterion level, and have modified Tables 3 and 2.4 so that the combination weights can be linked to the calculation of the RHI.

Point 7: Table 4 – Typo in the headers – Score Standard. The  table and text needs to make it clear that the table provides the scoring system for the ranges of combination weights for each index. Possibly the table could be reformatted to show this, it is not immediately apparent.

Response 7: Spelling errors have been corrected. The table is intended to provide a scoring system for the range of values assigned to each index and has been mentioned in 3.2.

Point 8: Table 8 – deals with the quantitative indices, it would be useful to have a similar table explaining how the more qualitative indices have been used and incorporated into the scoring system, otherwise it might be concluded that these are less important than the quantitative.

Response 8: We have modified Table 8 to include some qualitative indicators to complete the review.

Point 9: The conclusion in line 301 that the riparian condition index reflects the river health more than river water environment index more than  river morphology more than social service index could be explored a bit more, explaining why this might be and leading to a conclusion that a shortened version of this method could focus on riparian condition alone?

Response 9: We further explored the reasons for the greater importance of the riparian condition index in 4.1, but we do not believe it is straightforward to conclude with the available data and analysis that a shortened version of this approach could focus on riparian condition alone.

Reviewer 2 Report

This manuscript proposed a combination weighting method to assess river ecosystem health level, taking the Beijing section of Yongding River in China as a case study area. This work is valuable and significant to optimize the river development and management in a sustainable way. The results of this study are expected to provide important data for construction of water ecological environment locally. However, there are some problems with writing in this manuscript.

  Comments are as follows,

1. In the Introduction, some citations should be added to show crucial evidences for viewpoints of authors.

2. Reviews on methods of river health evaluation should be further expanded based previous studies. For instance, predictive models were widely utilized in many related studies. Additionally, the weight of indices are also measured by multiple methods such as Principal Component Analysis, and Criteria Importance Though Intercrieria Correlation. These methods should be supplemented in the reviews to show a comprehensive background of this study for readers.

3. In the Materials and Methods, the land use and land cover of this case study area should be depicted in the section 2.1. The detailed contents or indicators of the field survey should also be show to enhance the reliability and repeatability of this study.

4. Data sources of indicators for river health assessment should be depicted in detail.

5. The river connectivity should be considered as an indicator for evaluating river water environment.

6. The combined weights from geometric average method are likely to overestimate or underestimate the importance of certain indicators as shown in Table 3. That could induce a certainty of the assessment result of this study. How did the authors measure the reliability of the final weights? This point should also be explained in the Discussions.

7. The basis of the evaluation standards shown in Table 2 should be depicted to show its reasonability.

8. In the Results, how was healthy state from the analysis of the field survey result evaluated? This point is crucial to validate the reliability of the assessment results of this study.

9. Some statistical evidences, rather than qualitative descriptions should be showed in attribution analysis of the assessment results. For instance, in lines 240-242, statistics data on the planning and supervision of the government could provide a robust evidence.

10. In the Discussion, an uncertainty of this study caused by data being used should be analyzed in the section 4.1. Additionally, a comparison of this study with the related studies should be further supplemented to show the advantages and disadvantages of this work.

11. A broad implication of the case study should be highlighted to show the impacts of this study.

12. Line 67: the superscript of units should be checked carefully.

13. Please improve the expression of Figure 1. For instance, add the legend of elevations.

14. Please add a map scale in Figure 2.

Author Response

Response to Reviewer 2 Comments

Point 1: In the Introduction, some citations should be added to show crucial evidences for viewpoints of authors.

Response 1: We have added more citations in the introduction to testify the ideas of the article.

Point 2: Reviews on methods of river health evaluation should be further expanded based previous studies. For instance, predictive models were widely utilized in many related studies. Additionally, the weight of indices are also measured by multiple methods such as Principal Component Analysis, and Criteria Importance Though Intercrieria Correlation. These methods should be supplemented in the reviews to show a comprehensive background of this study for readers.

Response 2: Based on your comments, our review of river health assessment methods has been further expanded on previous studies.

Point 3: In the Materials and Methods, the land use and land cover of this case study area should be depicted in the section 2.1. The detailed contents or indicators of the field survey should also be show to enhance the reliability and repeatability of this study.

Response 3: We have changed the elevation data of the study area in Figure 1 to land use data, and added legends and scales, etc. Detailed survey indicators and methods we have added and modified in Table 1, thank you for your suggestions!

Point 4: Data sources of indicators for river health assessment should be depicted in detail.

Response 4: We have added in Table 1 a description of the qualitative indicators and the data sources.

Point 5: The river connectivity should be considered as an indicator for evaluating river water environment.

Response 5: After previous screening work, river connectivity is of low importance for river health evaluation in the region, so we did not use it as one of the indicators to evaluate the river water environment.

Point 6: The combined weights from geometric average method are likely to overestimate or underestimate the importance of certain indicators as shown in Table 3. That could induce a certainty of the assessment result of this study. How did the authors measure the reliability of the final weights? This point should also be explained in the Discussions.

Response 6: The method for calculating the combined weights is discussed further in 4.1.

Point 7: The basis of the evaluation standards shown in Table 2 should be depicted to show its reasonability.

Response 7: Spelling errors have been corrected. The table is intended to provide a scoring system for the range of values assigned to each index and has been mentioned in 3.2.

Point 8: In the Results, how was healthy state from the analysis of the field survey result evaluated? This point is crucial to validate the reliability of the assessment results of this study.

Response 8: The evaluation criteria in Table 2 are described in 2.4.

Point 9: Some statistical evidences, rather than qualitative descriptions should be showed in attribution analysis of the assessment results. For instance, in lines 240-242, statistics data on the planning and supervision of the government could provide a robust evidence.

Response 9: Additional statistics have been added to the attribution analysis of the assessment results.

Point 10: In the Discussion, an uncertainty of this study caused by data being used should be analyzed in the section 4.1. Additionally, a comparison of this study with the related studies should be further supplemented to show the advantages and disadvantages of this work.

Response 10: We further discuss the advantages and disadvantages of this work in terms of the use of data, etc., in 4.1.

Point 11: A broad implication of the case study should be highlighted to show the impacts of this study.

Response 11: We have further highlighted the implications and significance of this study in 4.1.

Point 12: Line 67: the superscript of units should be checked carefully.

Response 12: We have corrected the superscript of the units here and checked the superscript throughout.

Point 13: Please improve the expression of Figure 1. For instance, add the legend of elevations.

Response 13: We have improved the expression of Figure 1 and changed the legend to land use.

Point 14: Please add a map scale in Figure 2.

Response 14: We have expression the representation of Figure 2 by adding a scale and the labeling of important locations.

Round 2

Reviewer 2 Report

All suggestions have been carefully addressed.